# Enhancing Fine-Tuning Performance of Large-Scale Text-to-Image Models on Specialized Datasets

## Abstract

Fine-tuning pre-trained large-scale text-to-image models on specialized datasets has gained popularity for downstream image generation tasks. However, direct fine-tuning of Stable-Diffusion on such datasets often falls short of yielding satisfactory outcomes. To delve into the underlying reasons, we introduce a novel perspective to investigate the intrinsic factors impacting fine-tuning outcomes. We identify that the limitations in fine-tuning stem from an inability to effectively improve text-image alignment and reduce text-image alignment drift. To tackle this issue, we leverage the powerful optimization capabilities of contrastive learning for feature distribution. By explicitly refining text feature representations during generation, we enhance text-image alignment and minimize the alignment drift, thereby improving the fine-tuning performance on specialized datasets. Our approach is plug-and-play, resource-efficient, and seamlessly integrates with existing controllable generation methods. Experimental results demonstrate a significant enhancement in fine-tuning performance achieved by our method.

## 1 Introduction

In recent years, fine-tuning models pre-trained on large-scale datasets has become the research paradigm for downstream tasks (Raffel et al., 2020; Chowdhery et al., 2022; Zhang et al., 2022; Hoffmann et al., 2022). Image generation is no exception, with newly developed large-scale text-to-image models (e.g., Stable-Diffusion (Rombach et al., 2022), GLIDE (Nichol et al., 2021), DALLE-2 (Ramesh et al., 2022)) demonstrating impressive performance. Among these, Stable-Diffusion stands out as a widely used, open-source, and arguably one of the most effective large-scale text-to-image models.

When faced with a variety of text-to-image downstream scenarios, especially on specialized datasets (Wah et al.; Nilsback & Zisserman, 2008) (e.g., buildings of distinct styles, unique sub-species, specific individual faces), researchers often fine-tune Stable-Diffusion to achieve generation effects that better capture the domain's characteristics (Ruiz et al., 2023; Gal et al., 2022). However, the performance in practical applications doesn't always meet our expectations. In many cases, even when optimizing the text encoder concurrently, directly fine-tuning the Stable-Diffusion on these datasets fails to produce satisfactory results (see Figure 1). It cannot accurately generate data distribution and sample features consistent with the target dataset.

A natural approach is to provide more detailed control information, such as refined textual descriptions or more precise positional details. However, this strategy has inherent drawbacks. Using just text or positional information such as coordinates, dimensions, or scales often fails to encapsulate the full complexity and nuances of an image. Moreover, it escalates the annotation costs. As a result, the need for a more universal, straightforward, and efficient fine-tuning method has become pressing.

To address this, we delved deeply into the intrinsic factors affecting fine-tuning performance from the perspective of feature distribution. We introduced two indicators, text-image alignment and text-image alignment drift, and deeply analyzed them across different data and models. We found that although direct fine-tuning enhanced text-image alignment and reduced text-image alignment drift, the gains were relatively limited. This suboptimal performance can be attributed to the diffusion

model's indirect learning approach: rather than directly learning textual feature representations, it learns indirectly through predictions in the image space.

In light of this, we employed a contrastive learning approach (Radford et al., 2021), leveraging its powerful capability in feature distribution optimization. By directly optimizing textual feature representations during generation, we substantially enhanced the text-to-image alignment and reduced the text-to-image alignment drift. This improvement significantly boosted the fine-tuning performance on specialized datasets.

Our contributions are as follows:

- We conducted a quantitative analysis of the fine-tuning process in text-to-image generation models from a feature distribution perspective, uncovering the intrinsic factors affecting fine-tuning outcomes.

- We introduced a universal fine-tuning approach for text-to-image generation models tailored to specialized datasets. For scenarios emphasizing a specific niche or those with minimal inter-class variances—which are prevalent in downstream tasks—our model offers significant performance improvements.

- Our fine-tuning strategy is plug-and-play, demanding minimal additional resources and placing low requirements on the dataset.

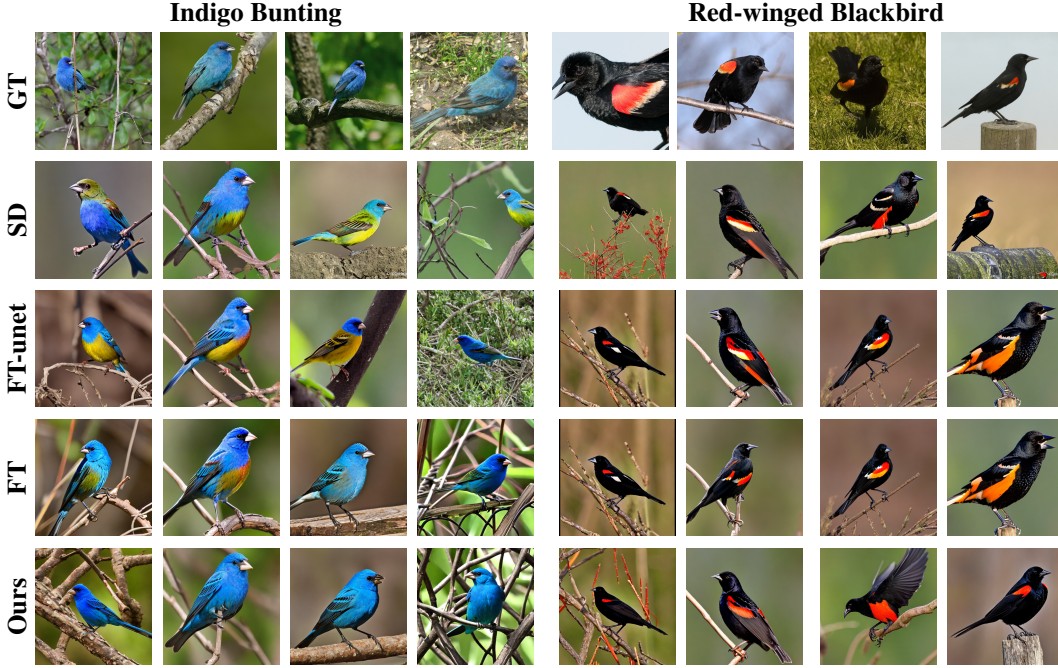

Figure 1: Images showcasing the Indigo Bunting and Red-winged Blackbird—two bird species from the CUB dataset. 'SD' is short for Stable-Diffusion, 'FT' is short for fine-tuning. 'FT-unet' implies that the text encoder is frozen, and only the unet is optimized. The generation results of pre-trained large-scale text-to-image models on these specialized datasets are often lacking, underscoring the need for fine-tuning. Traditional fine-tuning methods, whether focusing solely on the unet or jointly with the text encoder, fail to yield satisfactory outcomes. For illustration, while the Indigo Bunting is distinctively recognized by its vibrant blue hue, the produced image doesn't accurately reflect this characteristic color. Likewise, the emblematic red patches on the wings of the Red-winged Blackbird are misrepresented in terms of their placement by the fine-tuned model.

## 2 RELATED WORK

### 2.1 LARGE-SCALE TEXT-TO-IMAGE DIFFUSION MODELS

Building upon the ADM (Dhariwal & Nichol, 2021) architecture, GLIDE (Nichol et al., 2021) incorporated a text encoder and established a 3.5 billion parameter text-conditional diffusion model using classifier free guidance (Ho & Salimans, 2022). Its performance significantly surpasses previous diffusion models. Subsequently, Stable-Diffusion (Rombach et al., 2022) trains within the VAE's latent space and utilizes cross-attention to fuse text and image features, achieving highly realistic image generation. While models like DALLE2 (Ramesh et al., 2022) and Imagen (Saharia et al., 2022) also achieve comparable results, Stable-Diffusion, as an open-source model, has been widely adopted in downstream tasks and comes with a plethora of applications.

### 2.2 FINE-TUNING TEXT-TO-IMAGE MODEL

Since Stable-Diffusion is an open source model, most applications of text-to-image models are designed based on it. The Dreambooth method (Ruiz et al., 2023) has recently garnered significant attention in few shot image generation. Innovatively, Dreambooth uses rare tokens as unique identifiers, associates them with specific subjects, and fine-tunes the backbone and text encoder in Stable-Diffusion to synthesize relevant, photorealistic images. Notably, the Dreambooth method only addresses scenarios where the training set consists of a single object type. When the training set encompasses multiple categories or objects (for instance, the CUB dataset contains 200 bird species), Dreambooth requires training a separate generator for each category or object, which is highly inefficient.

In addition, Guo et al. (2023) achieve zero-shot image generation by learning the feature representation of text. Furthermore, a series of studies have explored various fine-tuning techniques with an emphasis on extending the model's functionalities (Zhang & Agrawala, 2023; Mou et al., 2023). These methods typically incorporate and optimize additional network structures while keeping the original text-to-image model frozen. This adaptation allows the model to accept alternative input forms, such as canny edge or user sketching.

Additionally, some research efforts have delved into fine-tuning diffusion models from the perspective of improving computational efficiency. For instance, the Low Rank Approximation (LORA) approach aims to alleviate the computational demands of training large-scale models and has been demonstrated to be effectively applied in the fine-tuning of text-to-image models.

## 3 IMPACT OF FINE-TUNING THE DIFFUSION MODEL ON FEATURE REPRESENTATIONS

In this section, we explore the impact of fine-tuning on Stable-Diffusion from the perspectives of text-image alignment and text-image alignment drift, both grounded in feature distribution analysis. Subsequently, leveraging these two indicators, we conducted a quantitative analysis on both real and generated data to evaluate the impact of fine-tuning on feature distributions. We choose CLIP as our multimodal feature extractor, which consists of a text encoder and an image encoder. Notably, the CLIP text encoder happens to be a component of Stable-Diffusion, therefore, fine-tuning Stable-Diffusion will also correspondingly alter the feature representation capability of the text encoder within it.

Throughout this paper, 'fine-tuning Stable-Diffusion' refers to jointly optimizing UNet and the text encoder, as opposed to solely optimizing UNet. Prior works indicate that joint optimization yields superior results, albeit with higher computational demands. This phenomenon is also illustrated in Figure 1.

### 3.1 TEXT-IMAGE ALIGNMENT

For the pre-trained CLIP model, the features extracted by the text encoder and image encoder from different modalities are aligned. The alignment of text and image features in the CLIP model enables a deeper semantic understanding between visuals and text, facilitates cross-modal tasks, and ensures

consistent and broad generalization across tasks. The contrastive loss function can be a indicator to assess this alignment between image and text data.

We found that the alignment of text and image features also affects the quality of generation (See Figure 2a). We used Stable-Diffusion to generate an image for each prompt on the CUB test set. Afterward, we randomly selected multiple categories and calculated the clip contrastive loss and FID for the generated results of each category. As can be observed from Figure2a, there is a clear positive correlation between FID and clip contrastive loss.

Then, we examine if fine-tuning can enhance the alignment between image and text data. For both the unfine-tuned and fine-tuned Stable-Diffusion, we extract the text encoder and pair it with the image encoder from CLIP to form a multimodal feature extractor, then use the contrastive loss function as a indicator to assess the alignment between text and image data. By randomly selecting ten batches of data, we compute the intra-batch contrastive loss.

The results are shown in Figure 2b. Given CLIP's extensive pre-trained on large datasets, there remains room for optimizing its alignment on specific downstream datasets like CUB. However, the decrease in fine-tuned Stable-Diffusion (green lines) is not significantly distinct from the unfine-tuned model (yellow lines). It means that although fine-tuning in greupdated the weights of the text encoder, there was no significant improvement in the text-image alignment.

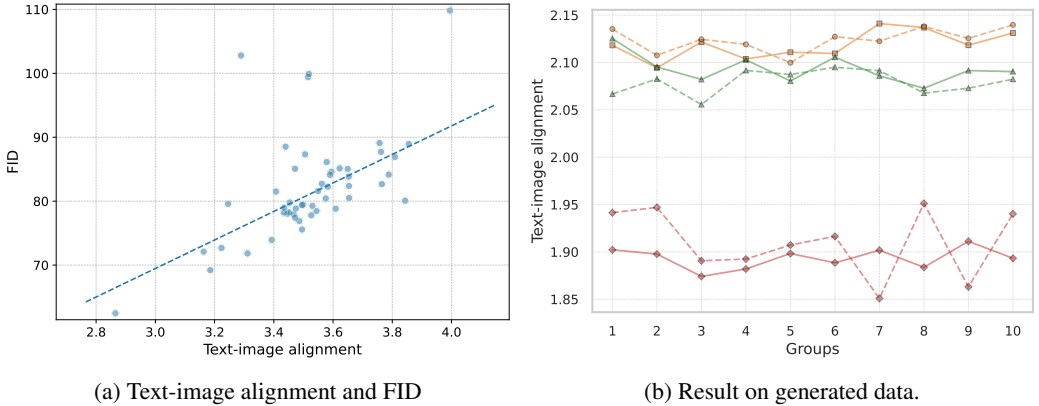

(a) Text-image alignment and FID          (b) Result on generated data.

Figure 2: **Left:** The relationship between text-image alignment and FID. The regression line was obtained using the method of least squares. **Right:** 'SD' is short for unfine-tuned 'Stable-Diffusion', 'SD-FT' is short for 'fine-tuned Stable-Diffusion'. For each model, we conducted an analysis based on the real data and its generated data.

**Conclusion:** Fine-tuning Stable-Diffusion doesn't significantly aid the text encoder in aligning text and image data.

## 3.2 TEXT-IMAGE ALIGNMENT DRIFT

After examining the alignment between the text feature $x$ and image feature $y$ for each model, we now turn our attention to the relationship between their respective changes, $\Delta x$ and $\Delta y$. We refer to this relationship as the *text-image alignment drift*. Consistent drift implies that altering the input text would consistently modify the generated image, a property beneficial for semantic interpolation (Song et al., 2020) and image editing (Hertz et al., 2022).

For each model, We still extract its text encoder and pair it with CLIP image encoder as a multimodal feature extractor. Then we randomly select two categories, and compute the Fréchet Distance between the text feature of the two categories as $\Delta x_i$, and the Fréchet Distance between the image feature of the two categories as $\Delta y_i$. After repeating the above process N times, we obtain the data $\{(\Delta x_i, \Delta y_i)\}_{i=1}^{N}$. We visualize the results using t-SNE and plot them in Figure 3.

To intuitively discern the data distribution across different models, we visualize each model's data separately using contour plots, as illustrated in Figure 4.

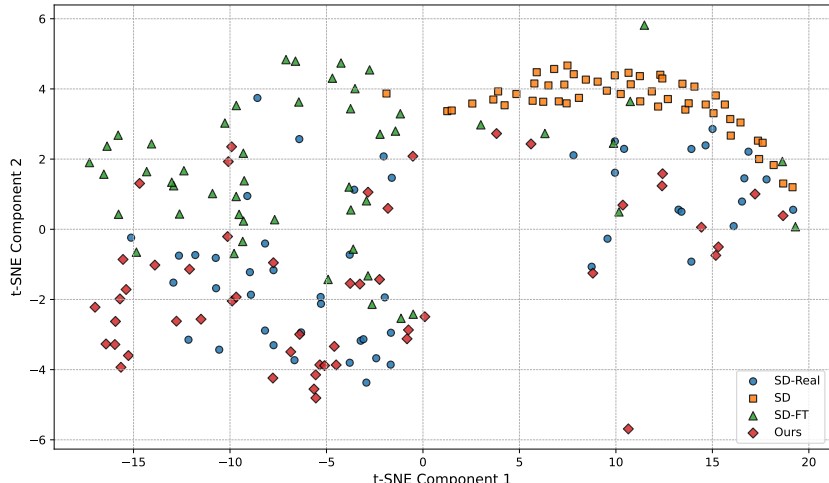

Figure 3: T-SNE result of feature alignment drift. **SD-real:** the unfine-tuned model on the real data. **SD:** the unfine-tuned model on corresponding generated data. **SD-FT:** the fine-tuned model on corresponding generated data. **Ours:** our model on corresponding generated data.

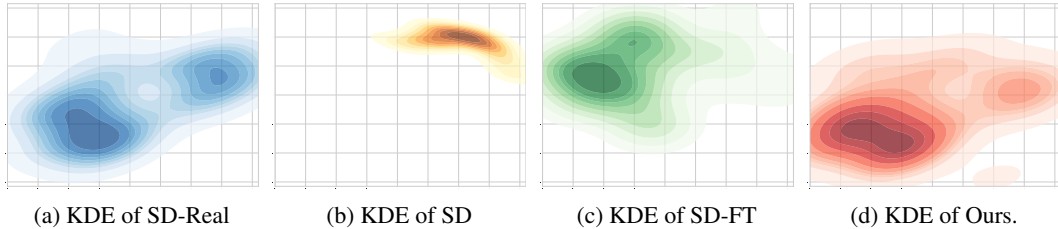

(a) KDE of SD-Real      (b) KDE of SD      (c) KDE of SD-FT      (d) KDE of Ours.

Figure 4: KDE of different models.

It's evident that the data of unfine-tuned Stable-Diffusion (yellow) is predominantly in the top-right corner, substantially deviating from the real data (blue). Even after fine-tuning, the data (green) remains considerably off from real data. This partially explains the challenge in fine-tuning the generated images by merely adjusting the text since the correlation between text and image alterations becomes disrupted. The red data points represent our method, elaborated in the subsequent section.

**Conclusion:** Fine-tuning struggles to capture the inherent alignment drift between text and image features.

## 4    CONTRASTIVE GENERATION

The analysis in the previous section revealed that direct fine-tuning on specialized datasets does not significantly improve the two indicators related to feature distribution that influence the generation effect. Therefore, during the training of our generative model, we fully exploit the ability of contrastive learning to optimize feature distributions, directly enhance the textual feature representation, rectify the two factors affecting the fine-tuning results, and thereby improve the fine-tuning performance.

Multimodal contrastive learning often operates on a sample-level basis. For image-text pairs $(x_i, c_i)$, we construct positive and negative samples $(x_i, c_j)$. When $i = j$, $x_i$ and $c_j$ are positive samples for each other, otherwise, they are negative samples for each other. However, in real-world scenarios, specialized datasets might only have category names without captions. Thus, besides integrating sample-level contrastive learning with the generation process, we also emulate scenarios with datasets lacking captions and introduce category-level contrastive learning where images and

texts within the same category mutually serve as positive samples. The overview of our approach is illustrated in Figure 5.

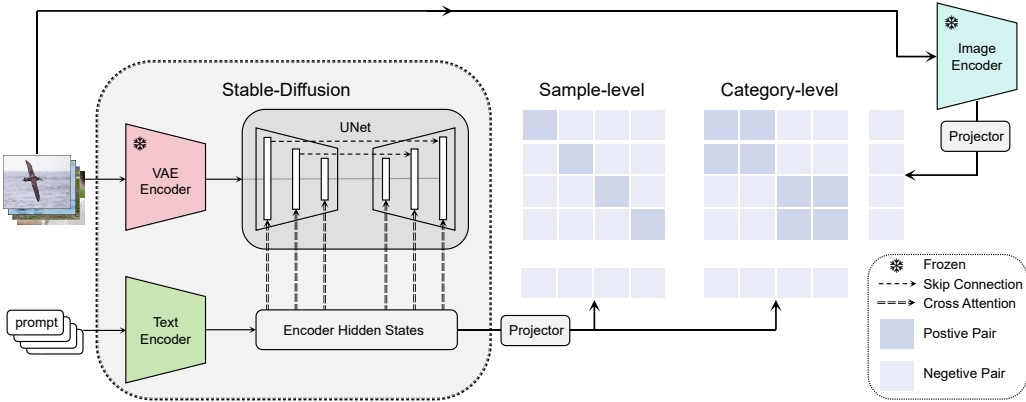

Figure 5: Overview of our approach.

**Prompt Design** A well-structured prompt format lays the foundation for optimizing textual features. Initially, the common prompt format for multimodal models was "a photo of [class]."(Radford et al., 2021). Additionally, in the design of prompts for diffusion models, the concept of an indicator has proven to be a successful approach (Ruiz et al., 2023). The primary idea is to utilize a specific token as a hint for a batch of data, allowing the model to learn the specific data set through that hint. However, for scenarios with numerous categories, it's challenging to allocate a rare token for each category simultaneously. Hence, we sought a more universal solution.

We turned our attention to the characteristics of the tokenizer's vocabulary. Tokenizers are widely recognized to exhibit three distinct characteristics: (i) Mapping low-frequency words to larger IDs; (ii) Assigning newly encountered tokens to newly generated IDs; and (iii) Avoiding the splitting of pure numbers. Given these insights, we adopted the strategy of using the category index incremented by 50,000 as the hint. This approach resonates with the peculiarities of specific datasets (such as face databases), wherein each entity or category is uniquely pinpointed by an index.

In summary, for datasets with captions, we employ the sample-level approach, and the prompt is designed as "[id of class] + 50000, a photo of [class], [caption]". for those without captions, we employ the category-level approach, the prompt is designed as "[id of class] + 50000, a photo of [class]".

**Sample-level Contrastive Generation** We use the bidirectional contrastive loss in CLIP model, which aimed at aligning image and text representations in a mutual embedding space. Given an image representation $v$ and a text representation $t$, the similarity between them is measured as:

$$\text{sim}(v, t) = \frac{v^\top t}{\|v\|_2 \cdot \|t\|_2}. \tag{1}$$

The objective of the loss is to augment the similarity between matching image-text pairs and diminish the similarity with negative samples. Formally, for a given image $v_i$ and its corresponding text description $t_i$, the loss considering both directions is:

$$L_{\text{ctr}}(v_i, t_i) = -\log \frac{\exp(\text{sim}(v_i, t_i)/\tau)}{\sum_{j=1}^N \exp(\text{sim}(v_i, t_j)/\tau)} - \log \frac{\exp(\text{sim}(v_i, t_i)/\tau)}{\sum_{j=1}^N \exp(\text{sim}(v_j, t_i)/\tau)}, \tag{2}$$

where $\tau$ is a temperature parameter and $N$ is the batch size.

**Category-level Contrastive Generation** At this point, the matching relationship between positive and negative samples has changed. Within the same category, images and texts are both considered as positive samples for each other. Accordingly, the contrastive loss is modified as follows:

$$L_{\text{ctr}}(v_i, t_i) = -\log \frac{\sum_{t_r \in P_i} \exp(\text{sim}(v_i, t_r)/\tau) + \sum_{v_s \in Q_i} \exp(\text{sim}(v_i, t_r)/\tau)}{\sum_{j=1}^{N} \exp(\text{sim}(v_i, t_j)/\tau)}$$
$$-\log \frac{\sum_{t_r \in P_i} \exp(\text{sim}(v_s, t_i)/\tau) + \sum_{v_s \in Q_i} \exp(\text{sim}(v_s, t_i)/\tau)}{\sum_{i=1}^{N} \exp(\text{sim}(v_i, t_j)/\tau)}, \tag{3}$$

where $P_i$ represents the set of texts in the batch that belong to the same category as the image $v_i$, and $Q_i$ represents the set of images in the batch that belong to the same category as the text $t_i$.

**Overall Optimization**   The total loss is:

$$L = L_{diff} + \lambda L_{ctr}, \tag{4}$$

where where $L_{diff}$ is the diffusion loss. The structure of our model is shown in Figure 3, where the image encoder is frozen and is used to guide the learning of the text encoder during the generation process.

## 5 EXPERIMENTAL

**Datasets**   Experiments are conducted on the CUB (Wah et al.) and Oxford Flowers (Nilsback & Zisserman, 2008) datasets. We consider two types of experiments: in the first, the original captions of the datasets are preserved, while in the second, we remove these captions, relying solely on category labels to mimic datasets used in downstream tasks without captions. The CUB dataset comprises images of 200 bird species, while the Oxford Flowers dataset covers 102 flower categories.

**Evaluation Metrics**   We report the widely-used Fréchet Inception Distance (FID) (Heusel et al., 2017) and Inception Score (IS) (Salimans et al., 2016) metrics. In addition, following (Sinha et al., 2021) we report the linear classification accuracy (Acc., measured in percentage) to more intuitively quantify the model's confusion regarding finer categorizations, which is an important metric for datasets like CUB, where the class differences are subtle. FID measures the distributional discrepancy between generated and real images, with a lower value indicating closer resemblance of generated images to real ones. IS evaluates the quality and diversity of generated images; a higher IS value typically signifies superior image quality and variety. Both metrics utilize the Inception network as a feature extractor. Some literature (Ye et al., 2023; Zhang et al., 2018) suggests that since the Inception network is trained on ImageNet, its IS metric might not be optimal for evaluations on other datasets. Therefore, for the CUB and Oxford Flowers datasets, we follow Zhang et al. (2018), using an Inception model fine-tuned on these datasets for feature extraction.

**Experimental Details**   We use Stable-Diffusion-2.1-base as our base model. The CLIP model we adopt is CLIP-ViT-H-14. We replace the text encoder without projection layers in SD with a text encoder that has projection layers. The image encoder also has a projection layer. Adam optimizer is used to train the network with base learning rates of 1e-5 for the generator and 1e-6 for the text encoder. We use a batch size of 24 and train the model for only 5 epochs, which approximately takes 3 hours on a single A100 GPU for the CUB dataset. A contrast coefficient of 0.1 is employed. During testing, we randomly select prompts from the test set and generate images from them. Each image in test set might correspond to multiple text descriptions; in such cases, we randomly choose one. For the fine-tuning of the Inception network, following the method in (Ye et al., 2023), we use the Adam optimizer with a learning rate of 1e-4, only fine-tuning the last layer.

### 5.1 COMPARISONS

**Text-image Alignment and Text-image Alignment Drift**   As Figure 2b shows, our method (red) directly enhances the alignment between text and image. As evident from Figure 2a, this improved alignment facilitates the learning of the generator. Figure 3 and Figure 4 reveal that the text-image alignment drift of our model closely resembles the alignment drift observed in the real data. This could imply that when the text features change, the generation results from our model are more stable and controllable.

**Quantitative Results** For the captioned CUB and Oxford Flowers datasets, we utilize an instance-level contrastive generation approach. Conversely, for the non-captioned CUB* and Oxford Flowers* datasets, we employ a class-level contrastive generation technique. As seen in Table 2, we have significantly enhanced the quality of generated images. Additionally, the observed improvement in the Acc. metric is indicative of our model's enhanced capability to capture the unique features intrinsic to each subclass, allowing for more precise categorizations.

Table 1: Quantitative results on captioned datasets.

| Model | CUB | | | Oxford Flowers | | |
|---|---|---|---|---|---|---|
| | FID | IS | Acc | FID | IS | Acc |
| SD | 18.97 | 5.35 | 56.24 | 22.14 | 3.95 | 40.32 |
| SD-FT-unet | 13.64 | 5.75 | 59.71 | 16.88 | 4.27 | 42.81 |
| SD-FT | 12.83 | 5.82 | 60.02 | 16.12 | 4.31 | 42.99 |
| Ours | 10.51 | 5.98 | 63.91 | 14.02 | 4.42 | 46.07 |

Table 2: Quantitative results on non-captioned datasets.

| Model | CUB* | | | Oxford Flowers* | | |
|---|---|---|---|---|---|---|
| | FID | IS | Acc | FID | IS | Acc |
| SD | 18.16 | 5.47 | 57.81 | 21.50 | 4.10 | 41.08 |
| SD-FT-unet | 14.55 | 5.70 | 59.40 | 17.72 | 4.26 | 42.77 |
| SD-FT | 14.04 | 5.75 | 59.94 | 17.24 | 4.29 | 42.83 |
| Ours | 12.15 | 5.81 | 64.02 | 15.50 | 4.36 | 46.75 |

**Qualitative Results** As Figure 1 shows (the bottom row), for the class 'Indigo Bunting', it is characterized by its blue hue, yet only the bird generated by our method match this coloration. Similarly, for the class 'Red-winged Blackbird', only specific areas on its wings manifest in red. Our model approximates the correct position of the red-wing, whereas other models do not.

In fact, we believe this result is notably significant, as we did not provide any additional detailed annotations (e.g., the position of the red-wing). Yet, the generated outcome shows more refined improvements. This might be attributed to our model's ability to better capture the feature distribution of the text during fine-tuning, leveraging the contrastive approach, and fully harnessing the potent capabilities of the pre-trained model.

## 5.2 ABLATION STUDY

For the multi-objective loss function, we analyze the impact of the balancing coefficient $\lambda$ on CUB datasets.

Table 3: Ablation study on $\lambda$.

| $\lambda$ | FID | IS | Acc |
|---|---|---|---|
| 0.75 | 12.14 | 5.80 | 63.87 |
| 0.1 | 12.15 | 5.81 | 64.02 |
| 0.3 | 15.27 | 5.48 | 60.05 |
| 1.0 | 20.72 | 4.88 | 45.45 |

## 6 CONCLUSION

In order to investigate the factors affecting the fine-tuning performance of text-to-image models on specialized datasets, we conducted quantitative experimental analysis from the perspective of

feature distribution. We identified text-image alignment and text-image alignment drift as indicators influencing fine-tuning outcomes. Based on these findings, we employed contrastive learning to directly optimize text features, leading to improvements in both text-image alignment and text-image alignment drift, thereby enhancing fine-tuning quality.

Given that Stable-Diffusion is not only the sole open-source large-scale text-to-image model but also arguably the most effective image generation model available, we conducted our experiments exclusively on Stable-Diffusion. However, since the text encoder is an essential component of text-to-image models, our method is also applicable to fine-tuning other architectures of text-to-image models.

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
