# OpenReview forum: "Enhancing Fine-Tuning Performance of Large-Scale Text-to-Image Models on Specialized Datasets"
_ICLR.cc/2024/Conference — Submitted to ICLR 2024_

### Official Review · Reviewer_mXUe · 2023-10-24

**Soundness:** 3 good
**Presentation:** 3 good
**Contribution:** 2 fair
**Rating:** 3
**Confidence:** 5

**Summary:**

This paper primarily investigates the fine-tuning of large-scale text-to-image generation models on specialized datasets. The authors found that while direct fine-tuning of the Stable-Diffusion model – a widely used, open-source, large-scale text-to-image model – on these datasets can enhance text-image alignment and reduce text-image alignment drift, the results were not satisfactory. This is mainly due to the indirect learning approach of the diffusion model, which learns indirectly through predictions in the image space, rather than directly learning textual feature representations.

To address this issue, the authors introduced a contrastive learning approach, directly optimizing textual feature representations during generation. This approach significantly improved text-to-image alignment, reduced text-to-image alignment drift, and greatly enhanced fine-tuning performance on specialized datasets.

**Strengths:**

The paper is well-written, starting with an analysis of the phenomenon, followed by the validation of the proposed method's improvement on existing tuning.

**Weaknesses:**

I believe the biggest problem with this paper is its novelty and experiments. The authors only conducted experiments on two real-world fine-grained datasets, which makes it difficult to demonstrate the broad improvements of the proposed method. Therefore, I recommend the authors to conduct experiments on more datasets, such as ood101, SUN397, DF-20M mini, Caltech101, CUB-200-2011, ArtBench-10, Oxford Flowers, and Stanford Cars. Furthermore, the community tends to use stable diffusion for artistic creation and rarely generates real photos nowadays. The proposed method improves contrastive learning for CLIP, which leans towards real photos. However, for a large amount of artistic creation, such as anime, character portraits, etc., I question the improvement of the proposed method. Please also prove the improvements of the proposed method on abstract datasets. Given your method's reliance on CLIP, I don't believe this will bring about significant improvements.

**Questions:**

please refer to the weaknesses

---

### Official Review · Reviewer_AZtN · 2023-10-25

**Soundness:** 2 fair
**Presentation:** 2 fair
**Contribution:** 2 fair
**Rating:** 3
**Confidence:** 4

**Summary:**

This paper provides a perspective to analyze the fine-tuned quality of text-to-image generation models. It finds that direct fine-tuning enhances text-image alignment very little, resulting in unsatisfactory generation results. And accordingly, this paper proposes to add an image-text contrastive loss to improve the fine-tuning and show some enhancement in two specific datasets.

**Strengths:**

The method proposed in this paper is easy to understand and follow. Experiments on two specific datasets also provide some demonstrations to prove its' effectiveness.

**Weaknesses:**

1. My main concern is the motivation of this paper. The paper strengthens the importance of text-image alignment in text-to-image generation but only uses the contrastive loss function (CLIP image and text features) as an indicator. As we know, image-text similarity calculated by CLIP cannot faithfully convey the exact image-text alignment, especially for some fine-grained features such as number, color, and relationship between objects. Can optimizing image-text contrastive loss result in good generation results? I can see only two cases in Figure 1.

2. Another concern is that the image-text pairs in the real world are noisy and intrinsically not fully matched. Optimizing the contrastive loss may not lead to good-generation images. Besides, you don't consider other image generation metrics other than FID and IS, which are not highly correlated with generated image quality. So can you conduct experiments on open-domain images and show more cases and other metrics to demonstrate the effectiveness of your method?

3. In figure 2(b), there is no legend in the graph (no "SD" or "SD-FT" denoted in the figure), and what do solid and dash lines and different colors mean?

4. What does KDE mean in Figure 4? You don't provide clear explanations and clarifications.

5. In [1], the paper also proposes an auxiliary loss called "image-text matching guidance", which is similar to the motivation of this paper. But you don't mention it or discuss it.


[1] Li, Wei, et al., "Upainting: Unified text-to-image diffusion generation with cross-modal guidance." arXiv preprint arXiv:2210.16031 (2022).

**Questions:**

Please see the weakness above.

---

### Official Review · Reviewer_kPtw · 2023-11-01

**Soundness:** 1 poor
**Presentation:** 2 fair
**Contribution:** 3 good
**Rating:** 3
**Confidence:** 4

**Summary:**

The authors propose a novel perspective to investigate the factors affecting fine-tuning outcomes and identify limitations in text-image alignment and drift. They leverage contrastive learning to refine text feature representations during generation, enhancing text-image alignment and minimizing alignment drift, leading to improved fine-tuning performance on specialized datasets.

**Strengths:**

- The authors' analysis of CLIP characterization is very interesting, and focusing on the impact of CLIP characterization on the generated results is a good direction. Moreover, the authors have designed a series of persuasive experiments to demonstrate this point. I believe that these analyses have some guiding significance for future research.
- The improvement method proposed by the authors is very direct and simple. Supervision in the feature space is indeed necessary. The authors have verified its effectiveness on small datasets.

**Weaknesses:**

- The experimental part lacks persuasiveness. 1) The authors have chosen small datasets and have not validated the proposed method on large datasets, making it difficult to demonstrate the effectiveness of this method. 2) The authors only showed the generated results of two types of birds in the qualitative analysis and did not provide more data or visualization results for more categories of data. Based on the current visualization results, it is difficult to determine the effectiveness of the proposed method.
- The ablation experiments are not sufficient, the authors need to analyze the results. At the same time, the authors can add the results of new datasets.
- The authors designed two types of loss functions. One is category-based, and the other is sample-based, but the author only verified the effect of one of them (i.e., category-based).
- It is difficult to determine whether the text space and image space are sufficiently aligned using FID and Acc as evaluation criteria. Please introduce indicators such as CLIP Similarity.

**Questions:**

- What does the KDE mean?
- Why not use the BLIP to generate the caption for non-caption data?
- Figure 2b lacks legend.

---

### Official Review · Reviewer_6e1T · 2023-11-07

**Soundness:** 2 fair
**Presentation:** 2 fair
**Contribution:** 2 fair
**Rating:** 3
**Confidence:** 4

**Summary:**

The paper introduces a contrastive learning framework to finetune text-to-image diffusion model improving image-text alignment. They have experimented on CUB and Oxfordflowers dataset and obtained improved performance.

**Strengths:**

1. The motivation is clear and the paper is relatively well-written.
2. The idea seems to be intuitive and experiments support this.

**Weaknesses:**

1. The observations drawn in the paper seem to be well-known. The text-image alignment drift is basically real to synthetic image domain gap, which is well-known to the community and therefore nothing new or interesting.
2. The paper addresses the actual issue of domain adaptation, but has not mentioned anywhere. Also, there exists a similar line of works, which tackles this issue. I would suggest comparing and contrasting these methods.
       a. Gal et al. “StyleGAN-NADA: CLIP-Guided Domain Adaptation of Image Generators”, SIGGRAPH 2022.
       b. Kim el al. “DiffusionCLIP: Text-Guided Diffusion Models for Robust Image Manipulation”, CVPR 2022.
3. Limited analysis, comparison and ablation studies provided in the paper. I would suggest the authors take these comments into consideration. E.g., ablation of sample level, category level generation etc. are missing.
4. In Fig.2 (b), legends are missing.
5. Writing needs to be improved. E.g., ‘improving image-text alignment’ and ‘reducing image-text alignment drift’ is basically the same.

**Questions:**

See weakness


##### After Rebuttal ###########

Authors did not submit rebuttal, so I am lowering my score.

---

### Meta-Review · Area_Chair_4wbd · 2023-12-09

**Metareview:**

This paper was reviewed by four knowledgeable referees, whose initial main concerns included:
1. the incomplete positioning and missing comparisons w.r.t. previous work (6e1T, AZtN)
2. the validation which appeared unconvincing due to limited experimental evidence (small scale datasets and missing ablations and in-depth discussions) (6e1T, kPtw, mXUe)
3. the metrics used to evaluate the proposed approach which did not include text-image alignment metrics (kPtw)
4. the motivation for the proposed approach which was unclear (AZtN, mXUe)

Unfortunately there was no rebuttal. The AC agrees with the initial assessment of the reviewers and therefore recommends to reject. The AC encourages the authors to take into consideration the feedback provided by the reviewers to improve future iterations of their work.

**Justification For Why Not Higher Score:**

Incomplete positioning and comparisons w.r.t. prior art, unclear motivation for the proposed approach and unconvincing evaluation. No rebuttal to address these concerns.

**Justification For Why Not Lower Score:**

N/A

---

### Decision · Program_Chairs · 2024-01-16

Reject